# Transporter-Mediated Interactions Between Uremic Toxins and Drugs: A Hidden Driver of Toxicity in Chronic Kidney Disease

**DOI:** 10.3390/ijms26136328

**Published:** 2025-06-30

**Authors:** Pierre Spicher, François Brazier, Solène M. Laville, Sophie Liabeuf, Saïd Kamel, Maxime Culot, Sandra Bodeau

**Affiliations:** 1MP3CV Laboratory, UR7517, Jules Verne University of Picardie, F-80000 Amiens, France; pierre.spicher@u-picardie.fr (P.S.); brazier.francois@chu-amiens.fr (F.B.); laville.solene@chu-amiens.fr (S.M.L.); liabeuf.sophie@chu-amiens.fr (S.L.); kamel.said@chu-amiens.fr (S.K.); 2Blood-Brain Barrier Laboratory (LBHE), UR 2465, University of Artois, F-62300 Lens, France; maxime.culot@univ-artois.fr; 3Department of Nephrology, Amiens University Medical Center, F-80000 Amiens, France; 4Department of Clinical Pharmacology, Amiens University Medical Center, F-80000 Amiens, France; 5Department of Clinical Biochemistry, Amiens University Medical Center, F-80000 Amiens, France

**Keywords:** chronic kidney disease, indoxyl sulfate, para-cresyl sulfate, indole, p-cresol, organic anion transporter, solute carrier transporter, intestinal barrier, blood–brain barrier, renal barrier

## Abstract

Chronic kidney disease (CKD) is associated with the systemic accumulation of uremic toxins (UTs) due to impaired renal elimination. Among these, indoxyl sulfate (IS) and p-cresyl sulfate (PCS) are particularly challenging because of their high protein binding and limited removal by dialysis. In addition to renal excretion, the transport of IS and PCS, and their microbiota-derived precursors, indole and p-cresol, across key physiological barriers—the intestinal barrier, blood–brain barrier, and renal proximal tubule—critically influences their distribution and elimination. This review provides an overview of transporter-mediated mechanisms involved in the disposition of IS, PCS, and their microbial precursors, indole and p-cresol. It also examines how these UTs may interact with commonly prescribed drugs in CKD, particularly those that share transporter pathways as substrates or inhibitors. These drug–toxin interactions may influence the pharmacokinetics and toxicity of IS and PCS, but remain poorly characterized and largely overlooked in clinical settings. A better understanding of these processes may guide future efforts to optimize pharmacotherapy and support more informed management of CKD patients, particularly in the context of polypharmacy.

## 1. Introduction

Chronic kidney disease (CKD) involves a progressive and irreversible decline in renal excretory function. As CKD advances, patients are affected by multiple comorbidities including cardiovascular, bone, metabolic, gastrointestinal, and neurological disorders [1,2,3,4,5,6]. These clinical manifestations have been associated with the accumulation of molecules in the blood and tissues that the kidneys would normally excrete. Among these molecules, uremic toxins (UTs) are considered to play a significant role in the pathophysiological mechanisms contributing to CKD-related complications [7]. Uremic toxins are commonly classified into three groups based on their physicochemical properties, molecular size, and removal efficiency during dialysis: (i) small water-soluble compounds, including substances < 500 Da such as urea and creatinine, which are easily dissolved in water and typically removed through dialysis; (ii) middle molecules, consisting of larger peptides and low molecular weight proteins (>500 Da), such as β2-microglobulin, which are less efficiently removed by conventional dialysis methods; and (iii) protein-bound uremic toxins (PBUTs), including compounds like p-cresyl sulfate (PCS) and indoxyl sulfate (IS), which bind to plasma proteins, limiting their clearance by dialysis. Notably, several PBUTs, such as PCS and IS, originate from the intestinal microbiota through the metabolism of dietary components [8]. Each group has distinct effects on the body and contributes to the complications associated with CKD progression as their concentrations rise in parallel with declining renal function, reaching their highest levels in patients with end-stage renal disease. Dialysis treatments, commonly initiated at this stage, effectively remove small UTs, and to a lesser extent, middle molecular weight compounds. However, dialysis has proven largely ineffective against PBUTs due to their high protein binding, particularly to albumin [9,10]. Consequently, these toxins accumulate in the blood and tissues, triggering uremic syndrome, which manifests as renal [7], cardiac [1,11], neurological [12], and bone disorders [13]. Among the PBUTs, IS and PCS exhibit the most thoroughly documented toxic effects [10]. These molecules originate from the metabolism of dietary amino acids: tryptophan for IS and tyrosine/phenylalanine for PCS [14,15]. In the intestine, bacterial tryptophanase converts tryptophan into indole, which enters the portal circulation and travels to the liver. There, cytochrome P450 2E1 (CYP2E1) and sulfotransferase 1A1 (SULT1A1) metabolize indole into IS [16]. Similarly, intestinal microbiota converts tyrosine and phenylalanine into p-cresol through tyrosine aminotransferase (TAT). The liver then processes p-cresol into PCS via sulfation by SULT1A1 [7]. Following their synthesis, IS and PCS bind extensively to plasma proteins (93% and 95%, respectively) [7,17,18]. These toxins circulate through the bloodstream, distribute to target tissues, and exert toxic effects, particularly on the neurological, cardiovascular, and renal systems. Ultimately, their elimination relies on proximal tubular secretion, as their high affinity for albumin prevents glomerular filtration [19].

While researchers have thoroughly studied the enzymatic pathways involved in the synthesis of IS, PCS, and their precursors, the transport mechanisms governing their distribution remain poorly understood. This review discusses the current knowledge on the transport mechanisms of two major UTs, IS and PCS as well as their microbiota-derived precursors (indole and p-cresol), across key physiological barriers: the intestinal barrier, the blood–brain barrier (BBB), and the renal barrier. These three sites are particularly relevant in the context of CKD, where alterations in toxin handling may contribute to their systemic or tissue accumulation and toxicity. We then examine potential interactions between these UTs and commonly prescribed drugs in CKD patients, focusing on shared transport pathways. Such interactions at the level of membrane transporters could modify the distribution and elimination of IS and PCS, potentially enhancing their toxic effects. This perspective highlights the clinical relevance of transporter-mediated drug–toxin interactions and suggests potential directions for improving medication strategies in CKD management.

## 2. Intestinal Passage of Indole and p-Cresol

While the systemic accumulation of IS and PCS in renal insufficiency is well-recognized, less attention has been given to the intestinal handling of their precursors, p-cresol and indole. Intestinal dysbiosis associated with CKD likely promotes the overproduction of these precursors, enhancing the generation of UTs. However, limited data are available regarding the absorption of p-cresol and indole within enterocytes prior to their hepatic conversion into PCS and IS. Below, we briefly review the prevailing hypotheses concerning this intestinal phase.

It is well-established that indole and p-cresol, once produced in the gut, are metabolized primarily in the liver by sulfotransferases and cytochrome P450 enzymes, which are highly expressed in this organ [8,20]. The hepatic synthesis of IS and PCS therefore implies that indole and p-cresol must enter the systemic circulation by crossing the intestinal barrier in their unmetabolized form.

Many transport proteins are expressed in the intestine [21,22,23] and could contribute to drug–toxin interactions or influence the development of CKD-associated comorbidities (Figure 1). Several studies have reported significant alterations in the expression and function of ATP-binding cassette (ABC) and solute carrier (SLC) transporters in pathological conditions including CKD [24]. In particular, the expression and activity of intestinal P-glycoprotein (P-gp or ABCB1) appear to be downregulated in uremic conditions, both in vivo and in vitro [25,26,27,28]. Similar reductions have been reported for other key transporters such as multidrug resistance-associated protein 2 (MRP2 or ABCC2) [26,29], MRP3 (ABCC3) [26], and organic anion transporting polypeptide 2B1 (OATP2B1 or SLCO2B1) [30]. The expression of breast cancer resistant protein (BCRP or ABCG2), another efflux transporter, has been shown to vary in CKD: some studies report overexpression [31,32], while others show decreased levels [25]. In contrast, Shimizu et al. observed upregulation of the peptide transporter 1 (PEPT1 or SLC15A1) in nephrectomized rats [33]. Such changes may not only alter the intestinal absorption of drugs and nutrients, but also influence the entry of microbial metabolites like indole and p-cresol into systemic circulation.

However, no study to date has demonstrated a direct role for these transporters in the translocation of indole or p-cresol across the intestinal barrier. As these molecules originate from microbial metabolism, most studies have focused on bacterial production pathways. Nevertheless, data from artificial membrane systems and bacterial models suggest that indole and p-cresol may cross lipid bilayers by passive diffusion [34,35]. This mechanism remains the most widely accepted route of intestinal absorption, although direct evidence in human intestinal cells is lacking [8,15,36,37,38].

Expanding on these findings, Soulage et al. recently proposed that the sulfation of p-cresol may occur directly within the intestine. They highlighted that SULT1A1, the enzyme responsible for the sulfation of both indole and p-cresol, is expressed at significantly higher levels in enterocytes than in hepatocytes [39]. This could account for the low concentrations of unconjugated p-cresol detected in the systemic circulation, suggesting that the molecule may enter the bloodstream predominantly in its sulfonated form. Furthermore, this hypothesis challenges the commonly accepted notion of the passive diffusion of indole and p-cresol across the intestinal epithelium [40]. In any case, the current knowledge remains insufficient to draw definitive conclusions regarding the contribution of this mechanism to the elevated IS and PCS levels observed in CKD patients.

Moreover, the interpretation of data on p-cresol concentrations in human plasma requires caution. Several older studies have reported the detection of circulating p-cresol, but many relied on non-specific or outdated analytical methods. As highlighted by Soulage et al., these techniques often failed to distinguish between unconjugated p-cresol and its sulfated form, pCS, or were subject to artifacts such as the spontaneous hydrolysis of pCS during sample handling. Consequently, the presence of p-cresol in plasma may have been overestimated or misinterpreted in earlier reports [41,42,43]. This should lead to a reconsideration of previous findings, as no robust evidence currently supports the presence of unconjugated p-cresol in the systemic circulation under either physiological or pathological conditions [40]. Future studies should rely on validated, specific analytical techniques to ensure accurate differentiation between p-cresol and its conjugates [44].

## 3. Transport of IS and PCS Across the BBB

During CKD, proximal tubular secretion of IS and PCS becomes impaired, resulting in the accumulation of these two uremic solutes in the systemic circulation [9,45]. The systemic and tissue accumulation of UTs contributes to multi-organ toxicity, with the central nervous system (CNS) being particularly vulnerable [12,46]. Several studies have reported the accumulation of IS and PCS in the brains of CKD animal models or in animals exposed to UTs [47,48,49,50]. These findings confirm that both compounds can cross the BBB and reach the CNS. Despite this, the molecular mechanisms underlying their passage across the BBB remain poorly understood, and only a few studies have explored the potential involvement of specific BBB transporters in IS and PCS trafficking. In this context, we examined the current literature on transporter expression and function at the BBB, and proposed additional hypotheses based on the existing knowledge of IS and PCS transport across other biological barriers such as the renal and intestinal epithelium. Understanding these mechanisms is essential for anticipating potential drug–UT interactions at the BBB. Such interactions could exacerbate tissue accumulation of IS and PCS within the CNS and intensify their neurotoxic effects.

The BBB expresses members of the SLC family and ABC transporters [51,52,53,54,55,56]. Since IS and PCS appear in the brain in the same chemical form as in the kidneys [57], it is plausible that the same transporters involved in their basolateral and apical handling across renal proximal tubular cells—namely OAT1 (organic anion transporter 1 or SLC22A6), OAT3 (organic anion transporter 3 or SLC22A8), and BCRP—also mediate their transport across the BBB [58,59,60]. Although OAT1 is not expressed at the BBB, OAT3 localizes to the basolateral membrane of brain endothelial cells and mediates the efflux of organic anions from the brain into the blood. Other SLC transporters may also participate in the CNS transport of UTs. For instance, OATP1A4 (SLCO1A4) and OATP2B1 contribute either to the entry of drugs into the brain at the basolateral side or to their efflux from endothelial cells at the apical membrane [53,61]. Among the ABC transporters expressed at the BBB, the BCRP efflux pump, located at the luminal membrane, exports drugs and xenobiotics unidirectionally from the intracellular compartment to the blood. MRP4 (ABCC4), meanwhile, shares a similar transport profile with OATP1A4 and OATP2B1, acting at both the entry and exit points depending on its membrane localization. Although these transporters are expressed at the BBB and are known to handle various organic anions and drug conjugates with structural similarities to certain UTs, there is currently no direct evidence that they mediate the transport of IS or PCS across the BBB [52,53,55,56,61]. Figure 2 summarizes the transporters that may be involved in the handling of IS and PCS within the CNS.

### 3.1. Transport of Indoxyl Sulfate (IS) Across the BBB 

Deguchi et al. were the first to demonstrate that IS exhibits greater efflux than influx across the BBB, using intracerebral microinjection, brain slice uptake assays, and in vivo integration plot analysis [62]. They reported a 2.6-fold higher clearance of IS from the brain compared to its entry, suggesting the existence of an active efflux mechanism. These findings are consistent with earlier observations of a saturable, carrier-mediated brain-to-blood transport of IS following intracerebral administration in rats. Notably, this efflux was significantly reduced in the presence of inhibitors known to target OAT3, such as probenecid, p-aminohippuric acid (PAH), benzylpenicillin, and cimetidine, whereas compounds primarily inhibiting other transporters, such as OAT1, OAT2, or OATP2 (e.g., salicylic acid, digoxin), did not significantly affect IS efflux. In addition, OAT3 mRNA expression was confirmed in rat brain capillary fractions. Supporting the role of OAT3 as an IS transporter, Ohtsuki et al. demonstrated an 11-fold higher uptake of IS in Xenopus oocytes expressing OAT3 compared with the water-injected controls [63]. Together, these data strongly support the involvement of OAT3 in the abluminal efflux of IS across the BBB in rats [64].

In addition, growing evidence confirms that CKD alters the expression of various ABC and SLC transporters not only in the kidney, but also in other tissues including the brain [65]. This naturally raises the question of whether IS, in addition to accumulating in the brain, might contribute to these changes in transporter expression. In this context, Burek et al. investigated the impact of IS exposure using an in vitro kidney–brain interaction model. This co-culture system, consisting of human brain microvascular endothelial cells (BMECs) and proximal tubular cells (HK-2), demonstrated the increased mRNA expression of several efflux transporters (P-gp, MRP1 [ABCC1], MRP4, MRP5 [ABCC5], and BCRP) following a 48-h exposure to 1 mM IS. Protein levels of MRP4 were also upregulated under these conditions. Similar upregulation was observed in vivo in brain capillaries isolated from mice subjected to bilateral renal ischemia/reperfusion injury [66]. Conversely, in a CKD model involving 5/6 nephrectomized rats, Naud et al. reported the decreased mRNA and protein expression of several transporters in brain capillaries including BCRP, MRP2, MRP3, MRP4, OAT3, OATP2, OATP3, and P-gp [67]. These divergent findings likely reflect the differences in the nature of renal injury—acute in the ischemia/reperfusion model versus chronic in the nephrectomy model. Nonetheless, collectively, these studies highlight a potential role of both SLC and ABC transporters in modulating IS transport across the BBB.

It is nevertheless important to interpret the above findings with caution. Most studies addressing the expression, regulation, and functionality of UT transporters at the BBB have been conducted in animal models. While these studies often demonstrate transporter expression through transcriptomic or proteomic analyses, the extent to which these data reflect the actual functional organization of the human BBB remains uncertain. Notably, Urich et al. reported substantial differences in the expression of ABC and SLC transporters when comparing in vitro models of the human BBB with mouse cerebral endothelial cells. These discrepancies were particularly evident for efflux pumps such as P-gp, MRP4, and BCRP as well as for OAT3, which was previously implicated in IS efflux by Ohtsuki et al. [63]. Importantly, these transporters appeared markedly overexpressed in mouse brain endothelium compared with in vitro human models [68], raising the question of whether such differences result from interspecies variability or reflect a downregulation phenomenon occurring during in vitro cell culture.

Recent transcriptomic analyses by Walchli et al., made accessible through the University of California, Santa Cruz (UCSC) Cell Browser [69], further complicate the interpretation of preclinical data by reporting no detectable expression of OAT1 or OAT3 in human BBB endothelial cells [70]. This finding reinforces the need for caution when extrapolating rodent data to human physiology. Furthermore, it is essential to consider that transporter expression may be influenced by disease states. Similar to observations in the intestine (see Section 2), the uremic environment associated with CKD has been shown to alter the expression of several transporters at the BBB in animal models [55,65,67]. While this has been clearly demonstrated in rodents, the extent to which such changes occur in humans with CKD remains unknown. Given that data on the baseline expression of UT transporters in the human brain are already scarce, it is currently not possible to confidently determine how these mechanisms are modulated during CKD. This underscores the need for further studies in relevant human models to better understand UT transport at the BBB in both health and disease.

### 3.2. Transport of Para-Cresyl Sulfate (PCS) Across the BBB

In the context of UT transport within the CNS, significantly more studies have investigated IS compared with PCS. Nevertheless, several reports have demonstrated elevated brain concentrations of PCS following intraperitoneal administration [48] or in mouse models of CKD [47], thereby confirming the existence of an influx of this UT into the CNS. However, it remains unclear whether the entry of PCS into the brain involves transporter-mediated mechanisms. To date, no studies have specifically addressed the interaction of PCS with transporter systems at the BBB. Notably, the accumulation of PCS in the CNS may be associated with its deleterious effects, potentially leading to the disruption of microvascular endothelial cells [50,71]. These findings highlight the need for further research to elucidate the mechanisms underlying PCS distribution in the CNS.

## 4. Renal Excretion of IS and PCS

In the kidneys, the elimination of IS and PCS occurs through the proximal tubule, as previously mentioned [45,72]. Since reduced elimination of these toxins is the primary cause of their accumulation in CKD, the excretion mechanisms of both toxins have been explored in recent years. Figure 3 illustrates the identified transporters involved in the renal uptake of IS and PCS.

### 4.1. Renal Excretion of Indoxyl Sulfate (IS)

Several studies have demonstrated that the SLC family, particularly the transporters OAT1 and OAT3, mediates the basolateral transport of IS across renal proximal tubular cells [45]. Enomoto et al. first demonstrated the involvement of rat OAT (rOAT1 and rOAT3) in IS tubular uptake. Through immunohistochemical analyses, they observed IS accumulation in the proximal and distal tubules of CKD rat models and confirmed the expression of rOAT1 and rOAT3 in proximal tubular cells. To confirm the role of these transporters in IS accumulation, they conducted in vitro experiments using proximal tubular epithelial cells stably expressing OAT1 and OAT3. Intracellular IS concentrations were reduced following exposure to OAT1 and OAT3 inhibitors such as probenecid and cilastatin [73]. Studies have indicated that benzbromarone, a hypouricemic agent that inhibits OAT1 and OAT3, decreases the total and renal plasma clearance of IS in rats by 34% and 43%, respectively [74]. Subsequent studies have explored how human OATs (hOATs) interact with IS using OAT-transfected proximal tubular and endothelial cells. These studies confirmed that IS is a substrate for both hOAT1 and hOAT3. They also demonstrated that inhibitors like probenecid and pravastatin significantly reduce the IS uptake mediated by these transporters [75,76,77]. Deguchi et al. validated Enomoto’s findings by investigating the transport of several UTs including IS, across porcine kidney 1 epithelial cells (LLC-PK1) and human embryonic kidney cells (HEK293) transfected with rOAT1/rOAT3 and hOAT1/hOAT3, respectively. To assess IS uptake, the authors cultured transfected cells on porous membranes and quantified the intracellular accumulation of radiolabeled IS. A significant increase in IS uptake was observed in transfected cells compared with non-transfected controls, and kinetic analyses supported a saturable transport mechanism mediated by rOAT1 and rOAT3. To extend their findings in a more physiologically relevant system, Deguchi et al. evaluated IS uptake in rat kidney slices preincubated with radiolabeled IS. By applying selective inhibitors of rOAT1 (PAH) and rOAT3 (pravastatin and benzylpenicillin), they estimated the respective contributions of these transporters and concluded that rOAT1 and rOAT3 play a comparable role in IS renal uptake [58]. Additionally, further studies showed that IS accumulates to a greater extent in the kidneys of CKD patients compared with non-CKD patients, particularly within tubular cells expressing OAT1 and OAT3 [78]. Knockout (KO) models of OATs have provided additional evidence. For instance, OAT1-KO mice exhibited elevated systemic IS concentrations [79]. Wu et al. observed similar results in OAT3-KO rats, where the plasma IS levels increased further in rats treated with probenecid. Using the OAT3-KO model, they estimated that OAT3 mediated approximately 75% of IS renal excretion [59]. Although this result highlights the major role of OAT3 in their experimental setting, it contrasts with the findings from Deguchi et al., who reported a more balanced contribution of OAT1 and OAT3 to IS uptake in rat kidney slices. IS has also been shown to inhibit OAT1- and OAT3-mediated transport of model substrates such as 6-carboxyfluorescein (6-CF) in vitro, supporting a potential for competitive interactions with other substrates [80]. This involvement was further supported by untargeted metabolomics in OAT1-KO mice, showing the plasma accumulation of IS, and by direct uptake assays in *Xenopus* oocytes expressing OAT1 [81].

Collectively, these studies establish the essential roles of OAT1 and OAT3 in mediating the basolateral renal excretion of IS in proximal tubular cells. Some studies suggest that OAT4 (SLC22A11) also contributes to IS transport [82]. Researchers observed higher IS uptake in human cells transfected with OAT4 compared with controls. This uptake decreased in the presence of OAT4 inhibitors like probenecid and pravastatin. According to these findings, OAT4 facilitates both IS influx and efflux at the apical membrane [76]. However, the literature provides limited data on IS transport by OAT4.

Other studies highlight how efflux pumps contribute to IS excretion from proximal tubular cells into the renal lumen. In 2018, Takada et al. reported higher IS plasma concentrations and lower survival rates in CKD mouse models compared with the controls, with these effects being more pronounced in BCRP-KO models. Using in vitro transport assays, they identified IS as a substrate of BCRP and demonstrated reduced IS excretion under BCRP-KO conditions [60]. In vivo experiments further showed that febuxostat, a potent BCRP inhibitor, leads to increased IS plasma concentrations in rats by impairing its excretion from proximal tubular cells [74]. Researchers also showed that IS inhibits the BCRP-mediated transport of estrone sulfate (ES) ammonium salt [83], underscoring the importance of BCRP in IS apical excretion. Regarding other ABC transporters, IS has been shown to inhibit the MRP4-mediated transport of methotrexate (MTX), indicating a potential interaction with MRP4; however, this does not necessarily imply that MRP4 is involved in IS transport itself [83]. Finally, the inhibition of BCRP and MRP4 in the presence of IS was associated with the decreased viability of conditionally immortalized proximal tubule epithelial cells (ciPTECs), likely due to the increased intracellular accumulation of IS and consequent cytotoxic effects such as oxidative stress and mitochondrial dysfunction [84].

### 4.2. Renal Excretion of Para-Cresyl Sulfate (PCS)

Since PCS is an anionic UT, similar to IS, both compounds share significant similarities in their handling by transporters. To investigate the basolateral uptake of PCS via OATs, Miyamoto et al. examined the effects of probenecid, IS, and other OAT1 and OAT3 inhibitors (PAH, benzylpenicillin, and ES) on PCS excretion using rat renal cortical slices and HK-2 cells. They reported a significant reduction in PCS uptake across all conditions, with benzylpenicillin and ES exerting stronger inhibitory effects than PAH. Based on these results, they concluded that both OAT1 and OAT3 mediate the basolateral renal uptake of PCS, with OAT3 playing a more prominent role [85]. Subsequent studies, however, nuanced these findings. The concentrations used in Miyamoto’s experiments (1 to 10 mM) greatly exceeded the known potencies of these inhibitors, suggesting a low affinity of PCS for both OAT1 and OAT3 and indicating potential cytotoxicity at these doses [86]. Nonetheless, lower concentrations of probenecid (1 µM to 2.5 mM) also reduced the PCS uptake in HEK293 cells expressing either hOAT1 or hOAT3 [77,87]. In line with these observations, PCS excretion was impaired in an OAT3-KO mouse model, resulting in increased systemic concentrations. Probenecid treatment further exacerbated this accumulation, supporting a roughly equal contribution of OAT1 and OAT3 in PCS elimination from the renal tubules, as probenecid selectively inhibits OAT1 in the absence of OAT3 [59]. André et al. further supported this transporter-mediated mechanism through a clinical study conducted in a cohort of patients with varying degrees of renal impairment. They demonstrated that the prescription of at least one OAT inhibitor was associated with increased plasma PCS concentrations. These in vivo findings align with and reinforce previous in vitro results, highlighting OATs as potential therapeutic targets for limiting UT accumulation and preventing the development of uremic syndrome [88].

Other transporters, such as OATP4C1 (SLCO4C1) [86] and OAT4 [82], have also been implicated in PCS handling, but no conclusive evidence has confirmed their contribution so far.

Mutsaers et al. clearly demonstrated BCRP’s role in PCS efflux, as they found enhanced PCS uptake in BCRP-expressing membrane vesicles and reduced PCS efflux in ciPTEC cells treated with the BCRP inhibitor KO143. They also reported that the highest PCS concentration tested (1 mM) inhibited the BCRP-mediated uptake of ES by 25%. In the same study, the authors investigated MRP4 and found no direct evidence that it transports PCS. Nevertheless, PCS inhibited MRP4-mediated MTX transport by approximately 40%, suggesting a possible interaction, without necessarily indicating that PCS is a direct substrate of MRP4 [89].

## 5. Drug–Transporter Interactions Affecting the Disposition of IS and PCS in CKD

Over the years, substantial evidence has established that UTs exert harmful effects independently. However, certain medications may exacerbate these deleterious effects by altering the pharmacokinetics of UTs. For instance, several drugs can impact the composition of the gut microbiota, thereby modulating the synthesis of precursors of gut-derived PBUTs [90,91,92,93]. PBUTs predominantly circulate in the albumin-bound form and can interact with other highly protein-bound drugs [94]. Theoretically, some medications could directly interfere with the protein binding of UTs. Indeed, Tao et al. described an increased free fraction of IS in the presence of ibuprofen in uremic plasma during their rapid equilibrium dialysis assays. This effect was further exacerbated in the presence of furosemide. They also obtained similar results in their ex vivo whole blood hemodialysis model [95]. Moreover, Shi et al. used ultrafiltration to separate free and human serum albumin (HSA) bound fractions of IS and PCS in vitro. After preloading HSA with IS and PCS, they observed that furosemide, indomethacin, warfarin, and ibuprofen significantly increased the free fraction of IS, while ibuprofen, warfarin, and furosemide exhibited a similar impact on PCS [96]. In addition, transporter-mediated interactions may also contribute to the exacerbation of uremic toxicity in CKD. The transporters involved in the disposition of PBUTs are known to handle a wide range of substrates including numerous drugs. Since these compounds often share common transport pathways, competition and inhibition may occur, potentially leading to increased blood and tissue concentrations of UTs [97]. In the following section, we discuss commonly prescribed drugs that influence the distribution and elimination of IS and PCS, with a particular focus on transporters located at the BBB and within the renal proximal tubule.

### 5.1. Polypharmacy in CKD

It is well-established that several conditions, such as diabetes or hypertension, represent major risk factors for CKD. Moreover, the progressive decline in kidney function contributes to the development of multiple comorbidities including anemia, cardiovascular disease, and mineral and bone disorders [98]. In this context, polypharmacy is highly prevalent among CKD patients, as reported in numerous cohort studies. The CKD-REIN (Chronic Kidney Disease—Renal Epidemiology and Information Network) cohort recently examined drug prescriptions in 3011 CKD patients recruited from 40 nephrology units across France. Among them, 2182 patients (72%) received six or more medications, with a median of eight drugs per patient [99]. As expected, the most commonly prescribed therapeutic classes were cardiovascular agents and antidiabetic medications. Several other cohort studies have investigated medication use in CKD populations in recent years. Table 1 summarizes the most frequently prescribed drugs in CKD patients, based on selected studies [99,100,101,102,103].

In this context, the increasing number of prescriptions further elevates the risk of competitive interactions with UTs, particularly with ABC and SLC transporters [97].

This list of frequently prescribed drugs provides a foundation for the next section, in which we examine whether existing data support potential interactions between these treatments and UTs, particularly through shared transporter pathways.

### 5.2. Impact of Prescribed Medications on IS and PCS Distribution and Elimination

Table 2 summarizes the data on the reported effects of various medications on IS and PCS transport across the kidney and BBB. This overview highlights treatments commonly prescribed for CKD, along with several widely used therapeutic classes including anti-infective agents and anti-inflammatory molecules.

For certain compounds, both in vivo and in vitro data support their potential to interact with IS and PCS. For example, probenecid, benzylpenicillin, cimetidine, and pravastatin have been shown to inhibit OAT-mediated transport of PBUTs. Since IS and PCS are well-established substrates of renal OATs, such inhibition can lead to increased systemic concentrations of these toxins by reducing their renal elimination [59,73,74,76,77,84,85,87,108]. Moreover, these compounds may also contribute to the accumulation of IS in the CNS by inhibiting its efflux via OAT3 at the BBB [63]. In addition, Taniguchi et al. reported that febuxostat, another drug for gout, promotes IS accumulation within the renal tubules of rats, likely through the inhibition of the BCRP transporter at the apical membrane of proximal tubular epithelial cells, thereby impairing IS efflux [74].

In 2002, Ohtsuki et al. demonstrated a significant inhibitory effect of ketoprofen on IS uptake in OAT3-expressing oocytes [63]. Given that several non-steroidal anti-inflammatory drugs (NSAIDs) are known substrates or inhibitors of OATs, Yu et al. subsequently investigated the impact of diclofenac, ketoprofen, and salicylic acid on the pharmacokinetics of IS in rats. Their results showed that both diclofenac and ketoprofen markedly increased the mean residence time of IS while reducing its renal clearance. Using Chinese hamster ovary cells (CHO) and HEK293 cell lines expressing hOAT1 and hOAT3, they further confirmed the inhibitory activity of these NSAIDs on both transporters, providing a mechanistic explanation for the observed reduction in IS clearance [105].

Building on these findings, they extended their investigations to cephalosporin and fluoroquinolone antibiotics. Ciprofloxacin was shown to inhibit OAT3, resulting in a similar alteration in IS pharmacokinetics [106]. Although Ohtsuki et al. had previously reported that cefazolin reduced the IS uptake in OAT3-expressing oocytes [63], Luo et al. did not observe any significant effect of cefazolin on IS pharmacokinetics in vivo [106]. More recently, Lu et al. demonstrated that proton pump inhibitors (PPIs) also act as potent OAT inhibitors, with both omeprazole and lansoprazole significantly reducing IS uptake in hOAT1- and hOAT3-transfected cells [104]. Regarding antihypertensive agents, Fujita et al. reported that quinapril, an angiotensin-converting enzyme (ACE) inhibitor, inhibited OAT3 activity and increased the serum IS levels in rats, while simultaneously reducing its renal clearance [107].

Since OATs remain the most extensively studied transporters for IS and PCS, it is not surprising that most available research on drug–toxin interactions has focused on these carriers. However, some of these compounds no longer hold significant therapeutic relevance. For instance, probenecid, although commonly used as a reference inhibitor of OATs in experimental settings, is now contraindicated in CKD. Likewise, benzylpenicillin has a very limited role within the current anti-infective arsenal. H2 receptor antagonists such as cimetidine are also prescribed far less frequently than PPIs. Similarly, febuxostat is generally not considered a first-line treatment, and the use of NSAIDs is contraindicated in patients with advanced CKD.

Although current knowledge about the interactions between CKD treatments and UTs remains limited, substantial evidence indicates that most therapeutic classes commonly prescribed in CKD act either as substrates or as inhibitors of ABC and SLC transporters. It is therefore reasonable to hypothesize that these drugs may influence the kinetics of PBUTs, and it would be highly relevant to investigate their impact on the distribution and elimination of IS and PCS. Table 3 provides a non-exhaustive list of these compounds.

Analyzing the impact of the therapeutic classes of interest in CKD on the transport of IS and PCS could provide clinicians with valuable insights to better understand and anticipate how CKD management strategies may affect the plasma and tissue concentrations of these UTs. Such knowledge may contribute to optimizing therapeutic strategies and improving patient outcomes in CKD management. Indeed, certain drug–toxin interactions may increase the plasma concentrations of these toxins, enhance their distribution to specific tissues, and ultimately exacerbate their toxicity.

## 6. Final Considerations

A better understanding of the transport mechanisms involved in the handling of UTs may contribute to improving the management of CKD. Among the UTs, PBUTs such as IS and PCS are particularly problematic due to their poor clearance by dialysis and broad tissue toxicity. Transporters from the SLC and ABC families play a key role in their systemic distribution and renal excretion. However, CKD alters the expression and activity of many of these transporters, potentially contributing to the systemic and tissue accumulation of these toxins. In this context, characterizing transporter-mediated pathways is critical—not only to understand the progression of uremic toxicity, but also to anticipate and manage potential drug–toxin interactions. Many commonly prescribed drugs in CKD patients are substrates or inhibitors of these transporters, raising the risk of competitive inhibition that could impair PBUT clearance or enhance their distribution to vulnerable tissues such as the brain or kidney. Improved knowledge of the impact of these medications on IS and PCS transport could help clinicians optimize pharmacotherapy and better manage UT burden. Future therapeutic strategies could focus on modulating transporter activity to enhance toxin elimination or restrict tissue distribution. Such approaches may complement current dialysis-based methods, which, despite ongoing advances, still show limitations in effectively removing PBUTs. Ultimately, integrating knowledge of transporter-mediated mechanisms into CKD treatment paradigms may help improve patient outcomes by minimizing drug–toxin interactions.

## Figures and Tables

**Figure 1 ijms-26-06328-f001:**
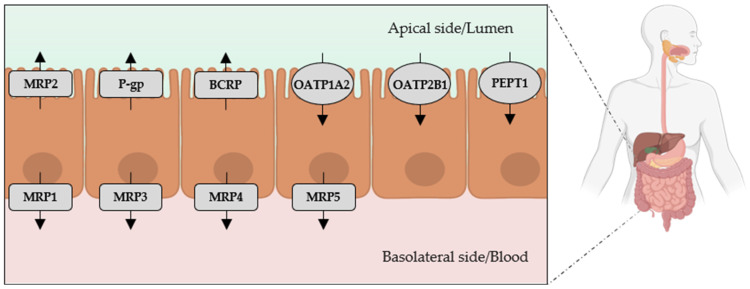
Expression and localization of intestinal transporters potentially altered in CKD: unclear implications for indole and p-cresol handling. BCRP: breast cancer resistance protein; MRP: multidrug resistance protein; OATP: organic anion transporting polypeptide; P-gp: P-glycoprotein; PEPT1: peptide transporter 1.

**Figure 2 ijms-26-06328-f002:**
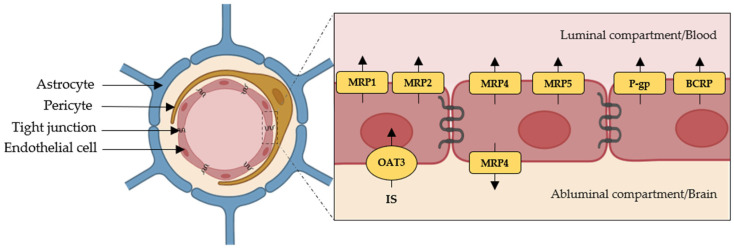
Expression and localization of candidate transporters potentially involved in indoxyl sulfate (IS) and p-cresyl sulfate (PCS) handling at the blood–brain barrier (BBB). BCRP: breast cancer resistance protein; MRP: multidrug resistance protein; OAT3: organic anion transporter 3; P-gp: P-glycoprotein.

**Figure 3 ijms-26-06328-f003:**
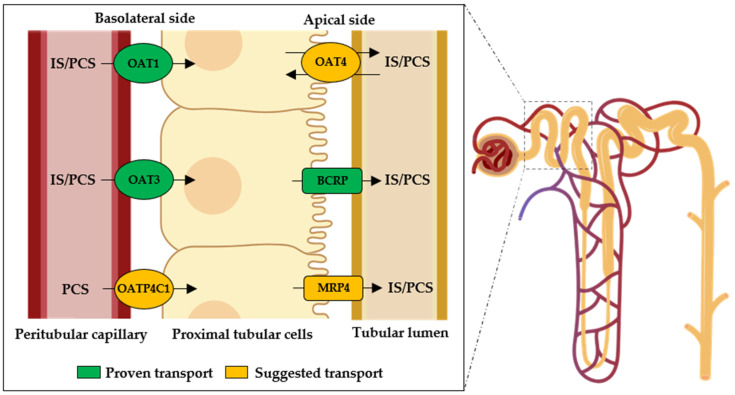
Renal tubular transporters involved in the secretion of indoxyl sulfate (IS) and p-cresyl sulfate (PCS). BCRP: breast cancer resistance protein; MRP: multidrug resistance protein; OAT: organic anion transporter; OATP: organic anion transporting polypeptide.

**Table 1 ijms-26-06328-t001:** Most frequently prescribed medications for CKD patients.

Therapeutics Class	Examples	References
Antianemics	Erythropoiesis-stimulating agents (EPO)	[100]
Anti-GERD agents	Proton pump inhibitors (PPIs)	[99]
Anticoagulants	Clopidogrel	[100,102]
	Direct oral anticoagulants (DOACs)	[100]
	Vitamin K antagonists (VKAs)	[99,100]
Antidiabetics	Dipeptidyl peptidase-4 inhibitors	[100]
	Glinides	[100]
	Insulin	[99,100]
	Metformin	[100]
	Sulfonylureas	[100]
Antigout agents	Allopurinol	[99,103]
Antihypertensives	Angiotensin-converting enzyme inhibitors	[99,100,101,102,103]
	Angiotensin II receptor blockers (ARBs)	[99,100,101,102]
	Beta blockers	[99,100,101,103]
	Calcium channel blockers	[99,100,101,102]
	Diuretics	[99,100,101,102,103]
CKD-MBD medications	Vitamin D	[99,103]
Lipid-lowering agents	Statins	[99,100,102,103]

CKD-MBD: chronic kidney disease-mineral and bone disorder; GERD: gastroesophageal reflux disease.

**Table 2 ijms-26-06328-t002:** Main therapeutic classes affecting IS and PCS central distribution and renal elimination.

Therapeutic Classes	Molecule	Observed Interactions with IS and PCS	Experimental System (Cell Line/Animal Model)	References
Anti-GERD	Cimetidine	OAT3 inhibition, increasing central concentrations of IS	In vivo (rat)	[63]
	Lansoprazole, omeprazole	OAT1 and OAT3-mediated IS transport inhibition	In vitro (MDCK-hOAT1/HEK293-hOAT3)	[104]
				[104]
Anti-inflammatory agents	Diclofenac	OAT1 and OAT3 inhibition, increasing mean residence time of IS while decreasing its renal clearance	In vitro (CHO-hOAT1/HEK293-hOAT3) In vivo (rat)	[105]
	Ketoprofen		In vitro (CHO-hOAT1/HEK293-hOAT3/ OAT3-oocytes) In vivo (rat)	[63,105]
Antibiotics	Benzylpenicillin	OAT3 inhibition, increasing central and systemic concentrations of IS or PCS	In vivo (rat)	[63,85]
	Cefazolin	OAT3-mediated transport inhibition of IS	In vitro (OAT3-oocytes)	[63]
	Ciprofloxacin	OAT3 inhibition, increasing area under curve and t1/2 of IS while decreasing its renal clearance	In vitro (HEK293-OAT3) In vivo (rat)	[106]
Antigout agents	Febuxostat	BCRP inhibition, inducing IS accumulation in the renal tubule	In vivo (rat)	[74]
	Probenecid	OAT1, OAT3 and OAT4 inhibition, increasing plasmatic concentrations of IS and PCS	In vivo (mouse/rat/human) In vitro (HUVEC, HK-2, S2-cells, HEK293-OAT1/ HEK293-OAT3) Ex vivo (Bioengineered kidney tubules)	[59,63,73,74,76,77,84,85,87,107,108]
Antihypertensives	Quinalapril	OAT3-mediated transport inhibition, increasing IS serum concentrations	In vivo (rat)	[107]
Antiviral	Acyclovir	OAT3-mediated transport inhibition of IS	In vitro (OAT3-oocytes)	[63]
Lipid-lowering agents	Pravastatin	OAT1, OAT3 and OAT4-mediated transport inhibition of IS	In vitro (S2-cells)	[76]

CHO: Chinese hamster ovary cells; GERD: gastroesophageal reflux disease; HEK293: human embryonic kidney cells; HK-2: human kidney 2 cells; HUVEC: human umbilical vein endothelial cell; MDCK: Madin–Darby canine kidney cells; S2-cells: immortalized cell line derived from the second segment of mice proximal tubule.

**Table 3 ijms-26-06328-t003:** Molecules commonly prescribed in CKD and identified as substrates or inhibitors of human, rat or mouse ABC, and SLC transporters.

Family/Therapeutic Class	Molecule	Inhibitor/Substrate	Transporter	References
Anti-GERD (PPIs)	Esomeprazole, rabeprazole	Inhibitor	OAT3	[109]
	Lansoprazole, omeprazole, pantoprazole		OAT1, OAT3	[109,110]
Antibiotics (β-lactams, fluoroquinolones, tetracyclines)	Cloxacillin, doxycycline, minocycline, nafcillin, oxytetracycline, piperacillin, tetracycline	Inhibitor	OAT1	[111,112,113]
	Cefalotin, ciprofloxacin		OAT3	[114,115]
	Cefamandole		OAT1, OAT3	[114]
	Amoxicillin, ceftibuten	Substrate	OAT1	[116,117]
	Cefaclor, cefdinir, cefoselis		OAT3	[117]
	Cefazoline, cefotiam, ceftizoxime, meropenem		OAT1, OAT3	[114,117,118,119,120]
	Tetracycline		OAT1, OAT3, OAT4	[112,113]
	Ampicilline, ceftazidime, piperacillin, tetracycline		MRP4	[111,112,113]
Antihypertensive (ACE inhibitors, ARBs, calcium channel blockers)	Captopril	Inhibitor	OAT1	[121,122,123,124,125,126]
	Olmesartan		OAT3	[127,128]
	Candesartan, enalapril, losartan, pratosartan, telmisartan, valsartan		OAT1, OAT3	[97,122,123,125,126,127,129,130]
	Verapamil		MRP4	[131]
	Captopril, olmesartan, quinalapril	Substrate	OAT1, OAT3	[113,121,122,123,124,125,126,127,128]
	Alacepril, enalapril		MRP4	[113,122,123,125,126,129]
Diuretics (loop diuretics, thiazide-like diuretics)	Bumetanide, chlorothiazide, cyclothiazide, trichloromethiazide	Inhibitor	OAT1, OAT3	[113,132,133,134,135,136,137]
	Hydrochlorothiazide		OAT1, OAT3, OAT4	[113,126,133,136,138]
	Furosemide		OAT1, OAT3, OAT4, MRP2	[97,113,126,133,134,136,137,138,139,140]
	Hydrochlorothiazide	Substrate	OAT1, MRP4	[113,126,133,136,138]
	Furosemide		OAT1, OAT3, MRP4	[97,113,126,133,134,136,137,138,139,140]
	Bumetanide		OAT1, OAT3, OAT4, MRP4	[113,132,133,134,135,136,137]
Lipid lowering agents (HMG-CoA reductase inhibitors, fibrates)	Atorvastatin, gemfibrozil, rosuvastatine	Inhibitor	OAT3	[141,142,143,144,145]
	Fluvastatin, simvastatin		OAT1, OAT3	[141,142,143]
	Pravastatin		OAT1, OAT3, OAT4	[76,113,141,142,143,144,146,147]
	Rosuvastatine	Substrate	OAT3	[141,142,143,144]
Lipid lowering agents (HMG-CoA reductase inhibitors, fibrates)	Pravastatin	Substrate	OAT3, MRP2, MRP4	[76,113,141,142,143,144,146,147]
	Bezafibrate		MRP4	[113]
Oral antidiabetics (sulfonylureas, glinides)	Chlorpropamide, nateglinide, tolbutamide	Inhibitor	OAT1	[148]
	Glibenclamide		OAT1, MRP2	[148,149]

HMG-CoA reductase: 3-hydroxy-3-methylglutaryl-coenzyme A reductase.

## Data Availability

Not applicable.

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
