# Peer review of "Transporter-Mediated Interactions Between Uremic Toxins and Drugs: A Hidden Driver of Toxicity in Chronic Kidney Disease"

_ijms, 2025, doi:10.3390/ijms26136328_

Round 1
Reviewer 1 Report
Comments and Suggestions for Authors
The manuscript presented by Spicher P. presents an overview of how Uremic Toxins are transported and their interactions in CKD.
The manuscript is well-written, but I have a few concerns.
- The manuscript needs an abbreviation list, and protein names must be described as first appearance.
- All figure legends need a better interpretation, including names for each figure.
- Some sections have very general information; please include more specific information in sections 2 and 3
- Lines 235 and 322, please explain these experiments better.
- Section 5.1 is just random information without a clear conclusion.
- Section 6.1, which includes plenty of clinical research, shows that this selection should be improved.
- Find the easy way to explain all the tables. They have a lot of unclear information, as shown in Table 3. (consider improving it).
- Be carfull do not duplicate information in different words
Author Response
Reviewer 1
- The manuscript needs an abbreviation list, and protein names must be described as first appearance.
As recommended, we have added a comprehensive list of abbreviations at the end of the revised manuscript (pages 24–25). Additionally, we have ensured that all protein names are spelled out in full upon their first appearance in the text.
We also took this opportunity to standardize the formatting of protein names throughout the manuscript (e.g., "organic anion transporter 1" with a single initial capital letter) for consistency and alignment with common scientific style conventions.
2.All figure legends need a better interpretation, including names for each figure.
Figure legends have been revised to include a clear title and a more informative description, in accordance with IJMS guidelines. Each legend now provides sufficient context to ensure the figure is understandable on its own.
3.Some sections have very general information; please include more specific information in sections 2 and 3
We acknowledge that some parts of Section 2 were initially general; however, we would like to emphasize that the information compiled on precursor molecules is, to our knowledge, not synthesized elsewhere in the literature. To enhance the section, we have added a paragraph discussing changes in the expression of intestinal transporters reported in patients with chronic kidney disease (CKD) (lines 114–128). This paragraph helps reinforce the physiological relevance of this barrier in CKD. In addition, we have restructured and reformulated the section to improve clarity and provide more focused content, in line with the reviewer’s suggestion. Additionally, we have included a paragraph warning against potential misinterpretations in the literature, where insufficiently specific analytical methods may have confused p-cresol with p-CS (lines 158–169).
In Section 3, we have added specific transcriptomic and proteomic data on transporter expression at the human BBB, highlighting discrepancies between species and experimental models. We also included recent findings showing that CKD alters the expression of several ABC and SLC transporters at the BBB, and discussed the limitations this imposes on the extrapolation of animal data to human physiology. These additions are particularly relevant given the limited knowledge currently available on UT transport across the human BBB (lines 248-274).
We believe these revisions significantly improve the specificity and scientific depth of the manuscript. The updated content can be found on pages 3-5 (Section 2) and 5-8 (Section 3).
4.Lines 235 and 322, please explain these experiments better.
In response to the reviewer’s suggestion, we revised the passages at lines 235 and 322 of the initial manuscript to provide clearer and more detailed descriptions of the cited experimental studies.
At line 235, the text now includes a clearer explanation of the experimental design and findings from Deguchi et al. The revised version describes the uptake assays performed using transfected cells and rat kidney slices, and how the authors used selective inhibitors (PAH for rOAT1 and pravastatin/benzylpenicillin for rOAT3) to estimate the contribution of each transporter. Additionally, we corrected a misstatement from the original text: the previously cited values of 38% and 62% corresponded to model predictions based on in vitro cell experiments, not measurements from kidney slices. This has been clarified in the new version (lines 318–328). To ensure consistency, we also revised a later sentence that incorrectly linked the 75% contribution estimated by Wu et al. to Deguchi et al.’s findings, which actually reported a more balanced involvement of rOAT1 and rOAT3 (lines 339-342).
In the same paragraph, we also corrected and clarified a separate passage originally located at lines 223–226. We replaced the incorrect mention of cisplatin with the correct drug cilastatin, and reworded the section (now lines 302–308) to improve clarity. Although not directly requested by the reviewer, this correction and rephrasing contribute to the overall clarification of the mechanisms discussed, in line with the reviewer’s suggestion.
At line 322, we have clarified the experimental approaches used to demonstrate drug–toxin displacement. Tao et al. employed rapid equilibrium dialysis to show that ibuprofen increases the free fraction of indoxyl sulfate (IS) in uraemic plasma, with an even stronger effect when furosemide was present. Additionally, we detail how Shi et al. used ultrafiltration to assess the displacement of IS and PCS from human serum albumin (HSA), demonstrating that furosemide, indomethacin, warfarin, and ibuprofen increased the free fraction of IS, and that ibuprofen, warfarin, and furosemide had a similar effect on PCS (lines 423-432).
5.Section 5.1 is just random information without a clear conclusion.
We agree with the reviewer that the original version of Section 5.1 lacked a clear purpose. We have revised the paragraph to better explain our rationale. The aim of this section is to identify the most frequently prescribed drugs in CKD patients as a starting point for our review. These medications were then used to investigate whether existing data suggest they could interact with uremic toxins (UTs), particularly through shared transporters, thereby potentially altering UT distribution or elimination and enhancing toxicity. This list provides the basis for the subsequent sections, which explore known or suspected transporter-mediated interactions. A concluding sentence has been added to clarify this intention.
Lines 460-462: “This list of frequently prescribed drugs provides a foundation for the next section, in which we examine whether existing data support potential interactions between these treatments and uremic toxins, particularly through shared transporter pathways.”
6.Section 6.1, which includes plenty of clinical research, shows that this selection should be improved.
As suggested by Reviewer 3 and in agreement with this comment, we have removed Section 6.1 from the revised manuscript.
7.Find the easy way to explain all the tables. They have a lot of unclear information, as shown in Table 3. (consider improving it).
We focused our efforts on improving Table 3, which has been reformatted and simplified to enhance clarity and readability. Minor adjustments were also made to the other tables to ensure consistency and improve comprehension where needed.
8.Be carfull do not duplicate information in different words
We have reviewed the manuscript to remove unnecessary repetitions and ensure the content is more concise.

Reviewer 2 Report
Comments and Suggestions for Authors
Spicher et al discuss about chronic kidney disease leads to the buildup of difficult-to-remove toxins like indoxyl sulfate (IS) and p-cresyl sulfate (PCS), which are strongly protein-bound. Their transport across key barriers and interactions with drugs affect their elimination but remain poorly understood. Better insight into these mechanisms could improve treatment and drug management in CKD patients.
Understanding how transporters handle uremic toxins (UTs), especially protein-bound toxins like indoxyl sulfate (IS) and p-cresyl sulfate (PCS), could improve chronic kidney disease (CKD) management. These toxins are poorly removed by dialysis and accumulate due to altered transporter function in CKD. Many drugs used in CKD can interfere with these transporters, increasing toxicity risks. Targeting transporter pathways may enhance toxin clearance and reduce tissue damage, complementing existing dialysis treatments.
The revision is coherent and accurately reflects the main arguments put forth by the authors
Author Response
We thank the reviewer for this positive assessment and are pleased that the revised manuscript clearly conveys our main arguments. No further changes were required based on this comment.

Reviewer 3 Report
Comments and Suggestions for Authors
In this review article the authors enlist the current knowledge on transporter mediated mechanisms involved in the handling of uremic toxins (UTs) such as indoxyl sulfate (IS) and p-cresyl sulfate (PCS). Further the review focuses on potential drug-drug interaction between UTs and commonly prescribed drugs in uremic toxins in chronic kidney disease (CKD) patients. Overall, the manuscript discusses the current knowledge gap in transporter mediated mechanisms in handling IS/PCS but makes an abrupt pivot to the role of IS/PCS in CKD disease progression. The latter part of the review does not fit well with the transporter mechanism sections and the reviewer recommends submitting these sections as two review articles for clarity and increasing reader-base for each topic.
Major Comments:
- The abstract needs to be re-worded, especially lines 22-28. For example, line 22 reads “This review synthesizes current knowledge….” And line 25-26, both these statements are not well worded and not clearly presented.
- Section 2 (Intestinal passage of indole and p-cresol), since the available literature is limited, the reviewer suggests moving section 2 into the Final Consideration part that summarizes the current Section 2. In that case Figure 1 may not be necessary.
- Figure 2, consider removing OATP1A4/2B1 from the figure as no data is available the depiction of OATP1A4/2B1 can be confusion to the readers.
- Section 4.1 (Indoxyl Sulfate), The authors based on the literature state that OAT1 and OAT3 would be the potential pathway for IS/PCS to enter the proximal tubule cells. But in line 224-226 line 224-226: it’s not clear from the summary of the study from reference 56 that how IS accumulation in proximal tubule cells of CKD rat models is observed if OAT1 or OAT3 is inhibited?
- One of the major comments the reviewer has is the quick pivot to the role of IS and PCS in CKD progression/disorders. This does shifts widely away from the transporter mediated IS/PCS DDI with CKD medications. It would be more impactful if the authors can consider having this as two different review articles than compressing multiple information in one review article.
Minor Comments:
- Abstract line 18, the field normally uses the term high protein binding instead of strong protein binding. Please make the necessary changes.
- Introduction line 76, “This narrative review aims to synthesize current….”, line 81 “ In a second step, we discuss…” Both these lines need to be rephrased better as reviews are not narrative but a discussion of the research topic from the perspective of the authors. As the word “perspective is used in line 90. Please make the necessary changes.
- Line 106: “the available literature generally suggests a high capacity…..” please rephrase this line or as mentioned in major comments #2 can be omitted if its not part of the summarized version.
- Line 152, “compounds into the brain at the basolateral…..” please rephrase the word compounds to drugs.
- Lines 246-253, these lines can be summarized in two sentences as the additional experimental details don’t add value to this section of the manuscript.
Author Response
Major Comments:
1.The abstract needs to be re-worded, especially lines 22-28. For example, line 22 reads “This review synthesizes current knowledge….” And line 25-26, both these statements are not well worded and not clearly presented.
We have reworded the indicated lines in the abstract to improve clarity and readability. The revised version provides a more concise and coherent summary of the review’s scope, with clearer language to describe the transporter-mediated mechanisms and their clinical relevance.
2.Section 2 (Intestinal passage of indole and p-cresol), since the available literature is limited, the reviewer suggests moving section 2 into the Final Consideration part that summarizes the current Section 2. In that case Figure 1 may not be necessary.
We acknowledge the reviewer’s suggestion and understand that the limited nature of the current literature may raise questions about the placement of Section 2. However, we believe that this very scarcity of data justifies the inclusion of a dedicated section. Providing a focused synthesis of existing knowledge and emphasizing current knowledge gaps is essential to support and encourage future research on the intestinal handling of indole and p-cresol.
As also suggested by Reviewer 1 (Comment 3), we have strengthened Section 2 by adding specific data on transporter expression changes observed in patients with chronic kidney disease (CKD), thereby improving the scientific relevance of the section (lines 114–128). Additionally, based on our own experience, we have included a paragraph warning against potential misinterpretations in the literature, where insufficiently specific analytical methods may have confused p-cresol with p-CS (lines 158–169).
We believe these additions enhance the relevance and clarity of Section 2 and justify its inclusion as a standalone part of the review. For this reason, we have also decided to retain Figure 1, which supports the section’s conceptual framework. To improve its clarity and alignment with the revised text, we have slightly modified Figure 1 by adjusting the size of transporter symbols and font, refining the legend, and adding the PEPT1 transporter.
3.Figure 2, consider removing OATP1A4/2B1 from the figure as no data is available the depiction of OATP1A4/2B1 can be confusion to the readers.
As suggested, we have removed OATP1A4/2B1 from Figure 2 to avoid potential confusion.
4.Section 4.1 (Indoxyl Sulfate), The authors based on the literature state that OAT1 and OAT3 would be the potential pathway for IS/PCS to enter the proximal tubule cells. But in line 224-226 line 224-226: it’s not clear from the summary of the study from reference 56 that how IS accumulation in proximal tubule cells of CKD rat models is observed if OAT1 or OAT3 is inhibited?
We have clarified the paragraph to distinguish in vivo and in vitro findings. Enomoto et al. observed IS accumulation in CKD rat tubules via immunohistochemistry, and subsequently confirmed OAT1/OAT3 involvement in vitro, where IS uptake was reduced in the presence of specific inhibitors. The revised text now reflects this more clearly.
Lines 300 to 308: “Enomoto et al. first demonstrated the involvement of rat OAT (rOAT1 and rOAT3) in IS tubular uptake. Through immunohistochemical analyses, they observed IS accumulation in the proximal and distal tubules of CKD rat models and confirmed the expression of rOAT1 and rOAT3 in proximal tubular cells. To confirm the role of these transporters in IS accumulation, they conducted in vitro experiments using proximal tubular epithelial cells stably expressing OAT1 and OAT3. HPLC measurements showed that intracellular IS concentrations were reduced following exposure to OAT1 and OAT3 inhibitors, such as probenecid and cilastatin”
5.One of the major comments the reviewer has is the quick pivot to the role of IS and PCS in CKD progression/disorders. This does shifts widely away from the transporter mediated IS/PCS DDI with CKD medications. It would be more impactful if the authors can consider having this as two different review articles than compressing multiple information in one review article.
As suggested by the reviewer, we have omitted Section 6 from the manuscript to maintain a clearer and more focused narrative.
Accordingly, we have revised the final paragraph of the introduction (lines 91-100) to reflect this structural change and ensure consistency throughout the manuscript.
Minor Comments:
1.Abstract line 18, the field normally uses the term high protein binding instead of strong protein binding. Please make the necessary changes.
This has been corrected as suggested. The term “strong protein binding” has been replaced with “high protein binding” in the abstract.
2.Introduction line 76, “This narrative review aims to synthesize current….”, line 81 “ In a second step, we discuss…” Both these lines need to be rephrased better as reviews are not narrative but a discussion of the research topic from the perspective of the authors. As the word “perspective is used in line 90. Please make the necessary changes.
We agree with the reviewer’s comment and have rephrased the sentences to better reflect the purpose and structure of the review, avoiding the term “narrative” and improving stylistic consistency with the rest of the introduction.
3.Line 106: “the available literature generally suggests a high capacity…..” please rephrase this line or as mentioned in major comments #2 can be omitted if its not part of the summarized version.
As mentioned previously, we have chosen to retain Section 2. In line with the reviewer’s suggestion, we have rephrased the sentence to improve clarity and avoid overstating the evidence. The revised version more accurately reflects the limited but consistent findings suggesting passive diffusion of indole and p-cresol across the intestinal barrier:
Lines 138–141: “Nevertheless, data from artificial membrane systems and bacterial models suggest that indole and p-cresol may cross lipid bilayers by passive diffusion [34,35].”
More broadly, Section 2 has been revised in response to Major Comment #2, to clarify its scope, incorporate more specific content (e.g., transporter modulation in CKD), and better articulate the rationale for maintaining this section as an independent part of the manuscript.
4.Line 152, “compounds into the brain at the basolateral…..” please rephrase the word compounds to drugs.
Corrected as suggested. The word “compounds” has been replaced with “drugs” line 196 of the revised manuscript.
5. Lines 246-253, these lines can be summarized in two sentences as the additional experimental details don’t add value to this section of the manuscript.
We agree with the reviewer’s suggestion and have revised this section to summarize the key findings in two concise sentences, while preserving the essential information regarding the role of OAT1 and OAT3 in IS transport.
Line 342-348: “IS has also been shown to inhibit OAT1- and OAT3-mediated transport of model substrates such as 6-carboxyfluorescein (6-CF) in vitro, supporting a potential for competitive interactions with other substrates [80]. This involvement was further supported by untargeted metabolomics in OAT1-KO mice, showing plasma accumulation of IS, and by direct uptake assays in Xenopus oocytes expressing OAT1 [81]”.

Round 2
Reviewer 3 Report
Comments and Suggestions for Authors
The authors have addressed all the comments of the reviewer.